# A Self-Management Programme of Activity Coping and Education - SPACE for COPD(C) - in primary care: The protocol for a pragmatic trial

Claire LA Bourne,[1] Pratiksha Kanabar,[1] Katy Mitchell,[1] Sally Schreder,[1] Linzy Houchen-Wolloff,[1] M John G Bankart,[2] Lindsay Apps,[1] Stacey Hewitt,[1] Theresa Harvey-Dunstan,[1] Sally J Singh[1,3]

[1]Collaboration and Leadership for Applied Health Research and Care—East Midlands, Centre of Exercise and Rehabilitation Science, Respiratory Biomedical Research Unit, Glenfield hospital, Groby Road, Leicester, Leicestershire, UK
[2]Department of Primary Care and Health Sciences, Keele University, Keele, UK
[3]National Centre for Sport and Exercise Medicine, Loughborough University, Leicestershire, UK

**Correspondence to**
Dr Claire LA Bourne; Claire. Bourne@uhl-tr.nhs.uk

## ABSTRACT

**Introduction** National guidance for chronic obstructive pulmonary disease (COPD) suggests that self-management support be provided for patients. Our institution has developed a standardised, manual-based, supported self-management programme: Self-Management Programme of Activity Coping and Education (SPACE for COPD(C)). SPACE was previously piloted on a 1-2-1 basis, delivered by researchers, to individuals with COPD. Discussions with stakeholders highlighted considerable interest in delivering the SPACE for COPD(C) intervention as a group-based self-management programme facilitated by healthcare professionals (HCPs) in primary care settings. The study aims are to explore the feasibility, acceptability and efficacy for the intervention to be delivered and supported by HCPs and to examine whether group-based delivery of SPACE for COPD(C), with sustained support, improves patient outcomes following the SPACE for COPD(C) intervention.

**Methods and analysis** A prospective, multi-site, single-blinded randomised controlled trial (RCT) will be conducted, with follow-up at 6 and 9 months. Participants will be randomly assigned to either the control group (usual care) or intervention group (a six-session, group-based SPACE for COPD(C)self-management programme delivered over 5 months). The primary outcome is change in COPD assessment test at 6 months. A discussion session will be conducted with HCPs who deliver the intervention to discuss and gain insight into any potential facilitators/barriers to implementing the intervention in practice. Furthermore, we will conduct semi-structured focus groups with intervention participants to understand feasibility and acceptability. All qualitative data will be analysed thematically.

**Ethics and dissemination** The project has received a favourable opinion from South Hampshire B Research Ethics Committee, REC reference: 14/SC/1169 and full R&D approval from the University Hospitals of Leicester NHS Trust: 152408. Study results will be disseminated through appropriate peer-reviewed journals, national and international respiratory/physiotherapy conferences, via the Collaboration and Leadership in Applied Health Research and Care and through social media.

**Trial registration** ISRCTN17942821; pre-results.

### Strengths and limitations of this study

► The burden of chronic obstructive pulmonary disease (COPD) is significant to both the health service and the individual. Supported self-management is important, but options are limited for those with COPD. This study explores a group-based supported self-management programme for individuals with COPD.
► This is a pragmatic trial where the study intervention (a group-based self-management support intervention for people with COPD) will be delivered and supported by healthcare professionals in community settings. The study has been designed to align with how the intervention might be delivered in routine clinical practice.
► Our follow-up period is 3 months post-intervention. Unfortunately, due to funding constraints, we are unable to carry out a longer term follow-up.

## INTRODUCTION

Chronic obstructive pulmonary disease (COPD) is the third leading cause of death worldwide and is associated with considerable disability, impaired quality of life and high utilisation of healthcare resources.[1] Symptoms and manifestations of the disease can be modified by adopting appropriate health behaviours including, but not limited to, exercise, physical activity, smoking cessation, anxiety management, breathing control, medication adherence and exacerbation management.[2] Acknowledging the importance of the role of the patient in adopting these behaviours, there has been a shift in attitude from a traditional paternalistic model of care towards a more collaborative approach for chronic disease management. The National Health Service (NHS) Five Year Forward View's aim is for the NHS to become better at helping people to manage their own health by staying healthy,

making informed choices of treatment, managing conditions and avoiding complications.[3] Inevitably, the patient is predominantly responsible for administering their own care and making choices about health behaviours that will affect their outcomes. Self-management support aims to inform and support patients in making these choices. National and international guidelines for the management of COPD suggest that self-management support should be provided for people with COPD, though at present evidence for how and when that support should be delivered is less robust.[2]

Reports in the literature describe programmes that have targeted interventions for patients who have been hospitalised with a COPD-related admission, often with the primary ambition of reducing future admissions.[4] These studies have had little impact on readmission. Arguably, the offer of supported self-management should be offered earlier in the disease trajectory. Other COPD self-management programmes beyond the UK have been described in a stable population. Although the models of care delivered are quite heterogeneous,[5] with some programmes providing up to 2 years of weekly supervised exercise training and education.[3 6–8] The infrastructure and resources required to provide such comprehensive support means they are unlikely to be deliverable to the breadth of the COPD population in the UK. In order to address this, we previously developed and tested a new self-management programme that offered a 'light touch' approach so that it could be provided on a larger scale.

A Self-Management Programme of Activity Coping and Education—SPACE for COPD(C)—aims to support people with COPD in managing day-to-day tasks, minimise symptom burden, provoke health enhancing behaviour change and enhance emotional well-being. The programme is structured around the SPACE for COPD(C) manual, which combines both generic self-management skills and disease-specific tasks. Pilot testing assessed the feasibility and acceptability of the intervention to patients,[9] and a fully powered randomised controlled trial assessed the efficacy of the intervention in primary care,[10] powered for change in symptom burden measured by the self-reported Chronic Respiratory Questionnaire (CRQ-SR) dyspnoea domain at 6 months. In these studies, the SPACE for COPD(C) manual was introduced to patients during an initial consultation with a healthcare professional (HCP) (using motivational interviewing techniques), followed by two telephone calls during the subsequent 6 weeks. Secondary outcomes included other domains of the CRQ-SR, shuttle walking tests, disease knowledge, anxiety, depression, self-efficacy, smoking status and healthcare utilisation measured at baseline, 6 weeks and 6 months follow-up. Results demonstrated significant short-term improvements in CRQ-SR dyspnoea, anxiety, fatigue and emotion scores, exercise performance and disease knowledge. At 6 months, anxiety, exercise performance and smoking status outcomes remained significantly different between the intervention group and the usual care group, though

there was no between-group difference in change in CRQ-SR dyspnoea.

Implementation focused work carried out following these studies with HCPs in primary care and local Clinical Commissioning Groups (CCGs), demonstrated considerable interest in delivering the SPACE for COPD(C) intervention as a group-based intervention rather than on a one-to-one basis. The most common theoretical rationale underpinning delivery of group-based self-management support interventions is Social Cognitive Theory/Social Learning Theory.[11 12] Bandura's (1977, 1997) social learning theory posits that behaviour is influenced by beliefs about one's ability to perform a particular behaviour (self-efficacy expectations), beliefs about the effectiveness of the behaviour (eg, the advantages and disadvantages of performing this behaviour; outcome expectations) and learning through social observation (including social norms, social support or pressure and the behaviours of others). Peer support and use of other patients as role models are approaches grounded in this theory and directly applicable to group-based self-management support interventions.

Delivery of SPACE for COPD(C) as a group-based intervention allows for several face-to-face contacts between patients and HCPs over a number of sessions. These contacts could be spread out further across a longer period, which may be more successful in maintaining behaviour change. Furthermore, having earlier sessions closer together in time allows group cohesion to take place, an important factor in optimising group dynamics.[13]

The SPACE for COPD(C) intervention has also previously been delivered by a member of the research team rather than by existing clinical services. If the group-based intervention were to be implemented in primary care following the current study, importance would be placed on delivery by HCPs in a format that is feasible and acceptable to HCPs, health service providers and to patients. Understanding how this intervention can be delivered within existing health services and identifying key barriers and facilitators to its implementation is an important next step in the development of this complex intervention.

### Aims and objectives of the study
This study aims to:
1. Examine whether group-based delivery of SPACE for COPD(C), with sustained support, improves patient outcomes following the intervention compared with a control group;
2. Explore feasibility, acceptability and efficacy of the intervention to be delivered and supported by HCPs. This will be done by:

   a. Exploring HCP's experiences of delivering the intervention and identify any barriers to delivery in practice;

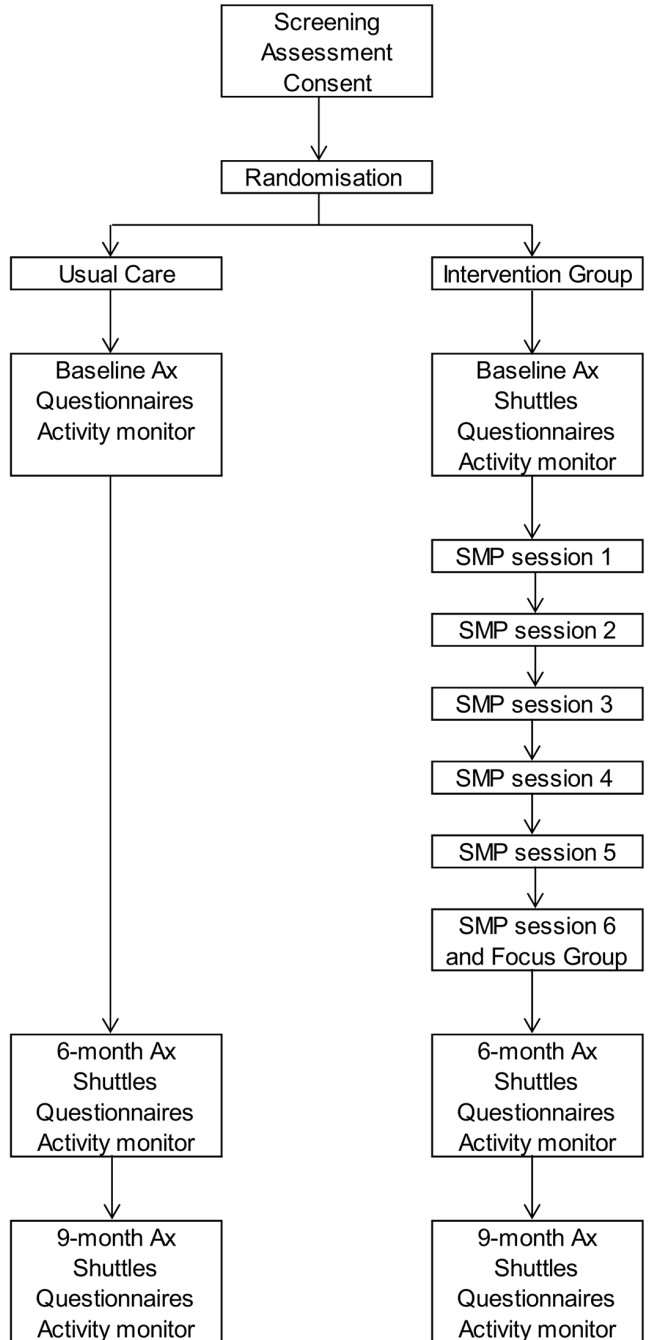

**Figure 1** Participant flow through the study.

b. Understanding, from the patient perspective, the feasibility and acceptability of the SPACE for COPD(C) intervention delivered by HCPs in a group-based community setting.

## METHODS/DESIGN
### Study design
The trial is a prospective, multi-site, single (assessor) blinded randomised controlled trial comparing a community-based, HCP-led, group-based self-management programme based on the SPACE for COPD(C) manual with usual care. The design of the study and flow

of participants is described in figure 1. The study will be run across Leicestershire and Rutland, and a total of 150 participants will be recruited (75 in the intervention group and 75 in the control group).

### Recruitment of participants
We will recruit participants with COPD, who will be identified from primary care (General Practice; GP) COPD registers and from patients who respond to a poster advertisement that will be displayed at GP practices and hospitals. We will also recruit participants from the following areas within the Respiratory Biomedical Research Unit at University Hospitals of Leicester:
► Those who have been involved in previous research trials who have agreed to be contacted again; or
► Those who were unsuitable for previous research trials but who agreed to be contacted about future research trials for which they might be eligible.

### Participant invitation
Eligible individuals identified as having an established diagnosis of COPD are sent an invitation letter, a patient information sheet about the study and a reply slip. For those recruited directly from primary care, the invitation letters are sent by the primary care practice where the search was conducted. For those recruited from existing databases, the invitation is sent from the principal investigator of the study. Individuals who are interested in taking part are asked to return a reply slip directly to the SPACE for COPD(C) research team. Interested participants are then contacted via their preferred contact method, and an appointment is arranged for a baseline visit.

### Eligibility criteria
Participants are eligible for the trial if they have:
► An established diagnosis of COPD as defined by The Global Initiative for Chronic Obstructive Lung Disease (GOLD) criteria.
Patients are excluded from participating in the trial if they are:
► Unable to participate in the exercise component of the SPACE for COPD(C) programme due to neurological, locomotive or psychiatric disability;
► Unable to participate in the exercise component of the SPACE for COPD(C) programme due to other comorbidities where exercise would be a contraindication (e.g., unstable angina);
► Unable to read/write English to the level of an 8 year old;
► Unwilling to be randomised;
► Previous participants of pulmonary rehabilitation or have received the SPACE for COPD(C) manual in the previous 12 months.

### Randomisation
Once participants have consented to take part in the study and spirometry has confirmed a COPD diagnosis, participants are randomised by an unblinded member

of the study team using an online randomisation tool (sealed envelope).[14] Individuals are randomised (1:1) to the control group or the intervention (SPACE for COPD(C) group-based self-management programme) group. The system randomises patients in random permuted blocks. This allows for the 1:1 ratio, but due to the random permuted blocks of 2, 4 or 6 ensures full randomisation. Simple randomisation has been chosen as there has been no requirement to stratify by age, gender, location or other variables. Participants are immediately informed of their allocated treatment by an unblinded member of the study team. Unblinding is permissible in the case of medical emergencies (eg, cardiac arrest) or patients being admitted to hospital for an exacerbation.

### Study interventions
#### Usual care (control group)

Participants in the control group will continue with any usual check-ups/reviews, and there will be no additional care provided or removed from their current access. If patients are referred to pulmonary rehabilitation in the duration of their time in the study, they will not be denied access to the programme; however, they will not be included in the final analysis due to the use of 'intention-to-treat' analysis. No additional advice, information or recommendations will be provided to participants in this group.

#### SPACE FOR COPD group-based self-management programme

Participants in the intervention group receive a SPACE for COPD(C) manual and are asked to attend the SPACE for COPD(C) group-based self-management programme usually within 1 month of their baseline appointment. The aim of the SPACE for COPD(C) programme is to support people with COPD in managing day-to-day tasks, minimise symptom burden, provoke health enhancing behaviour change and enhance emotional well-being. The programme is structured around the SPACE for COPD(C) manual, which combines both generic self-management skills and disease-specific tasks. The programme is facilitated by two trained HCPs (eg, physiotherapists, respiratory specialist nurses, occupational therapists and health psychologists) to groups of up to 10 participants and delivered through six 2-hour sessions, over a 5-month period. These sessions will be held at community venues, at times and locations to suit participants of the group to increase retention. Participants are liaised with in regards to preferences on timings and location of the group sessions to increase retention and engagement in the intervention. The content of the programme and accompanying self-management components[15] are presented in tables 1 and 2. Participants are provided with a contact number for at least one of the facilitators throughout the course of the programme in case they have any further queries/are unable to attend any sessions.

Participants will also be asked to complete the exercise component of the manual at home in their own time. A full description of the rationale, development and efficacy

**Table 1** SPACE for COPD(C) self-management programme session outline

| Session 1 (week 1) | Introduction to SPACE for COPD(C) |
|---|---|
| 1 | Welcome and introductions |
| 2 | Group responsibilities |
| 3 | What does it mean to have COPD? |
| 4 | What is self-management? |
| 5 | How to use the SPACE for COPD(C) manual |
| 6 | Goal setting |
| 7 | Home activities for next session |
| 8 | Summary and close |
| Session 2 (week 2) | Introducing exercise and managing shortness of breath |
| 1 | Welcome back |
| 2 | Solution focused goal feedback |
| 3 | Managing shortness of breath |
| 4 | Introduction to the walking programme |
| 5 | Goal setting |
| 6 | Activities for next session |
| 7 | Summary and close |
| Session 3 (week 4) | Continuing exercise and saving energy |
| 1 | Welcome back |
| 2 | Solution focused goal feedback |
| 3 | Saving your energy |
| 4 | Strength training |
| 5 | Goal setting |
| 6 | Home activities for next session |
| 7 | Group discussion |
| 8 | Summary and close |
| Session 4 (week 8) | Managing stress and emotions and the COPD action plan |
| 1 | Welcome back |
| 2 | Solution focused goal feedback |
| 3 | Managing stress and emotions |
| 4 | Action plans |
| 5 | Goal setting |
| 6 | Activities for next session |
| 7 | Summary and close |
| Session 5 (week 14) | Question and answer |
| 1 | Welcome back |
| 2 | Solution focused goal feedback |
| 3 | Question and answer |
| 4 | Goal setting |
| 5 | Activities for next session |
| 6 | Summary and close |

Continued

**Table 1** Continued

| Session 6 (week 20) | Keeping going from here |
|---|---|
| 1 | Welcome back |
| 2 | Solution focused goal feedback |
| 3 | Hobbies |
| 4 | Maintaining exercise |
| 5 | Sharing success |
| 6 | Summary and close |

COPD, chronic obstructive pulmonary disease; SPACE for COPD(C), Self-Management Programme of Activity Coping and Education.

of the work underpinning the SPACE for COPD(C) manual is detailed elsewhere.[9] The intervention will be offered over a period of 2 years (the duration of the active component of the intervention within the study) and will only be offered as part of the research, not routine practice within the community.

### Intervention fidelity

Intervention facilitators are registered health professionals (physiotherapists, respiratory nurse specialists and health psychologists). In total, eight registered health professionals will be trained to deliver the SPACE for COPD(C) group-based intervention by two health psychologists.

All facilitators attended a 1-day training course to ensure that they understood the theories and philosophy underpinning the SPACE for COPD(C) group-based programme and the content and resources used within it. All facilitators were given a facilitator manual to support their delivery of the programme and given the opportunity to practise delivering at least one activity from the manual during the training session.

### Quality assurance

Quality assurance will be undertaken to assess delivery of intervention content and educational style. Intervention fidelity checklists for intervention facilitators and trained observers have been specifically designed for the study. Intervention facilitators will complete checklists at the end of each self-management group session, and one of the trainers will observe one session per self-management group, completing their own checklist. Intervention facilitators will receive written and verbal feedback from the trained assessor.

### Study outcomes

Data will be collected during baseline, 6-month and 9-month appointments at the Leicester Respiratory Biomedical Research Unit by trained members of the study team. Data are collected following standardised operating procedures. Written informed consent is obtained from all participants prior to the commencement of data collection. Details of all clinical assessments and outcome measures are provided in table 3. General

practitioners are informed of patients' participation in the trial and any relevant results. Any serious adverse events will be reported to the sponsor and patients' ability to exercise safely will be monitored.

### Primary outcome

The primary outcome is health status, as measured by the COPD Assessment Test (CAT)[16] at 6 months post-baseline. This measure was chosen due to ease of use in clinical practice compared with the Chronic Respiratory Questionnaire (as used in the previous RCT). The CAT is a validated, short (eight items) and simple patient completed questionnaire and assesses globally the impact of COPD (cough, sputum, dyspnoea and chest tightness) on health status. The CAT is scored 0–5 with a range of 0–40; scores of 0–10, 11–20, 21–30, 31–40 represent mild, moderate, severe or very severe clinical impact, respectively.

### Secondary outcomes

#### Clinical measures

*Exercise capacity*

Maximal exercise capacity will be measured with the incremental shuttle walk test (ISWT) according to the protocol of Singh et al[17] using a 10 m course. According to current standards, an individual change of at least 47.5 m is considered clinically important.[18] Endurance capacity will be measured with the endurance shuttle walk test using a 10 m course and a walking speed of 85% of the maximal ISWT walking speed.[19]

#### Physical activity

Physical activity is assessed using physical activity monitors. The 'Bodymedia Sensewear' (APC Cardiovascular, UK) activity monitor is a biaxial accelerometer that can report a number of parameters including step count and energy expenditure. We will also use these data to assess compliance to the physical activity recommendation of undertaking at least 150 min of moderate intensity physical activity per week in bouts of at least 10 mins. Participants are asked to wear the activity monitor on the back of their right arm for seven consecutive days (24 hours a day if possible) following their baseline, 6-month and 9-month visits.

#### Questionnaires

*Health-related quality of life (HRQOL)*

HRQoL data will be measured using the European Quality of Life-5 Dimensions (EQ-5D[20 21]). The EQ-5D is a standardised questionnaire that was developed for use as a measure of health outcomes and defines health in terms of five dimensions: mobility, self-care, usual activities, pain or discomfort and anxiety or depression.

#### Self-reported Chronic Respiratory Questionnaire

Disease-specific HRQoL will be measured by the self-administered standardised CRQ-SR.[22] An individual change of at least 0.5/domain (dyspnoea, fatigue, emotional functioning and mastery) is considered clinically important.[23]

**Table 2** Taxonomy components present in the SPACE for COPD(C) facilitator manual, elaboration of the techniques under each component, direct examples from the SPACE for COPD(C) facilitator manual and the dose of the component

| Taxonomy component | Elaboration | Examples from the SPACE for COPD(C) facilitator manual (dose) |
|---|---|---|
| A2. Information about available resources | | Participants are provided with information throughout the programme (every session). |
| A3. Provision of/agreement on specific clinical action plans and/or rescue medication | | Session 4: Action plans (session 4 only). |
| A6. Practical support with adherence (medication or behavioural) | Walking and strength training diaries are provided for participants and discussed during sessions. | Walking and strength training diaries are provided for participants and discussed during solution focused goal feedback at the beginning of sessions 3–6 (sessions 3–6). |
| A8. Safety netting | Participants are able to call programme facilitators between sessions if needed | Participants are provided with contact details for programme facilitators who they can call if needed (this is a constant throughout the programme). |
| A11. Training/rehearsal for practical self-management activities | Including:<br>▶ managing shortness of breath<br>▶ saving your energy | Session 2: Managing shortness of breath (session 2 only).<br>Session 3: Saving your energy (session 3 only). |
| A12. Training/rehearsal in psychological strategies | Including:<br>▶ goal setting (including action planning)<br>▶ solution focused goal feedback<br>▶ problem solving<br>▶ self-reward and social reward<br>▶ managing stress and emotions | Goal setting activity (including action planning) and solution focused goal feedback (once every session).<br>Problem solving (this is a constant throughout the programme).<br>Session 4: Managing stress and emotions (session 4 only). |
| A13. Social support | Including:<br>▶ practical support<br>▶ emotional support | Participants are encouraged to share experiences, advice, ideas and support each other (this is a constant throughout the programme). |
| A14. Lifestyle advice and support | Including:<br>▶ introduction to the walking programme<br>▶ strength training<br>▶ hobbies<br>▶ maintaining exercise | Session 2: Introduction to the walking programme (session 2 only).<br>Session 3: Strength training (session 3 only).<br>Session 6: Hobbies (session 6 only).<br>Session 6: Maintaining exercise (session 6 only). |

SPACE for COPD(C), Self-management Programme of Activity Coping and Education.

**Table 3** Details of study clinical assessments and outcome measures at all appointments

| Baseline appointment (blinded and unblinded study team members) | 6-month appointment (blinded study team member) | 9-month appointment (blinded or unblinded study team member) |
|---|---|---|
| Consent<br>Collection of demographic details and medical history<br>Blood pressure<br>Spirometry<br>Randomisation*<br>Questionnaires (CAT, EQ-5D, CRQ-SR, BCKQ, PAM, HADS)*<br>Shuttle walking tests* (intervention participants only): 2xISWT; 1xESWT<br>Participants given Senswear activity monitor to wear for 7 days* | Check consent<br>Questionnaires (CAT, EQ-5D, CRQ-SR, BCKQ, PAM, HADS)<br>Shuttle walking tests†<br>Participants given Senswear activity monitor to wear for 7 days | Check consent<br>Questionnaires (CAT, EQ-5D, CRQ-SR, BCKQ, PAM, HADS)<br>Shuttle walking tests<br>Participants given Senswear activity monitor to wear for 7 days |

*Carried out by an unblinded member of the study team.
†ESWT carried out by an unblinded member of the study team.
BCKQ, Bristol COPD Knowledge Questionnaire; CAT, COPD Assessment Test; CRQ-SR, Chronic Respiratory Questionnaire; EQ-5D, European Quality of Life-5 Dimensions; ESWT, endurance shuttle walking test; ISWT, incremental shuttle walking test; HADS, Hospital Anxiety and Depression Questionnaire.; PAM, Patient Activation Measure.

There is both an initial and follow-up version depending on time of administration.

### Anxiety and depression

Depression and anxiety will be measured using the Hospital Anxiety and Depression (HADS) Scale to produce independent subscales for anxiety and depression.[24] The HADS is a self-report rating scale of 14 items on a 4-point Likert scale range (0–3). The HADS is a validated and a widely used questionnaire for screening for the separate dimensions of anxiety and depression and possible occurrence of anxiety and depression from patients[25] and the general population.[26] It measures anxiety and depression (seven items for each subscale). Cronbach's coefficient was 0.884, which indicates good reliability. Published cut-off scores for clinically relevant indications of depression and anxiety recommend a score of 8 for each subscale.[27]

### Patient activation

Patient activation (participants' knowledge, skill and confidence for managing their own health and healthcare) will be measured using the Patient Activation Measure.[28] This is a 13-item patient-reported measure that has been validated in the UK as a powerful and reliable measure of patient activation. Participants indicate their level of agreement on a four-point scale (strongly disagree to strongly agree) and responses are added to yield a raw score between 13 and 52. The raw score is calibrated to an activation score between 0 and 100 (the higher the score the higher the level of activation), which is then used to classify participants into one of four levels of activation (level 1: low activation; level 4: high activation).

### COPD knowledge

The Bristol COPD Knowledge Questionnaire will be used to understand patients' informational requirements and understanding and their knowledge base of COPD.[29] The questionnaire is comprehensive and goes into detail, regarding various aspects of COPD, for example epidemiology, signs and symptoms and exacerbations and treatment.

### Outcomes to assess feasibility and acceptability of trial parameters

We will use the following outcomes to assess the feasibility and acceptability of trial parameters:

### Screening

Defined as the number of packs sent out to patients from GP practices and assessed for eligibility using inclusion/exclusion criteria by a study researcher.

### Eligibility

Calculated by dividing number of people screened for eligibility by those who meet the inclusion criteria.

### Consent

Defined as the proportion of people with COPD who met inclusion criteria, and were therefore eligible, who went on to consent in writing to participate in the study.

### Group characteristics

Group characteristics (eg, age, gender, GOLD stage, Medical Reseach Council (MRC) dyspnoea grade, exercise capacity, physical activity) will be compared between completers and non-completers.

### Retention

Defined as the number of participants who remain in the study and do not drop-out.

### Study completion

Defined by the number of participants who complete the CAT at 6-months. Completion rates will be calculated at baseline, 6-month and 9-month follow-up.

### Intervention adherence and completion rates

This will be measured by summing the total number of self-management programme sessions attended by participants allocated to the intervention group. We will also look at the average group size across each of the

six self-management programme sessions and compare with the number of participants allocated to each of the self-management programme groups.

## Qualitative data collection

Those participants allocated to the intervention group are invited to take part in qualitative focus groups at the end of the SPACE for COPD(C) intervention. Focus groups have been chosen due to their generation of information on collective views and the meanings that lie behind those views. The aim of the focus groups will be to understand participants' experiences of the group-based self-management programme. More specifically, the data we collect will inform:

► Acceptability and usefulness of the programme to participants in this format and over this time period;
► The content of the intervention;
► Approaches to recruitment.

Focus groups will be conducted with each self-management programme group, with between 3 and 10 participants (number dependent on each group size). This difference in participant numbers allows for participant opinions to be gathered even if a small group is encountered (eg, due to drop-out). Although three is a very small number for a focus group, it allows all participant opinions to be gathered, regardless of group size. Participants will be familiar with one another (which can help facilitate discussion or the ability to challenge each other comfortably) as they have attended multiple group sessions together. Purposive sampling will be employed to recruit intervention participants. Audio-recorded focus group discussions (approximately 60 min) will be conducted face-to-face between each participant group, an experienced interviewer and an observer/note-taker. Focus groups will be carried out at the end of the last group session for participant ease, as discussed with study patient representatives. We will prompt participants allocated to each self-management group of the focus group discussion prior to the last session in the attempt to gain experiences from as many participants as possible, regardless of the number of sessions attended in total. Focus groups will be transcribed verbatim by a professional transcriber, with identifiable information removed. Focus group questions have been devised based on relevant literature and experience of the team.

HCPs delivering the SPACE for COPD self-management support intervention will be invited to participate in a meeting to discuss aspects of feasibility and acceptability, such as gaining insight into any potential facilitators/barriers to implementing the intervention in practice (and derive practical recommendations for doing so). Minutes will be taken during the discussion and anonymised.

## Data analysis
### Study power
The power calculation was based on the primary outcome at 6 months.[30][31] To detect a mean±SD between-group

difference of 2.5±5.0 in the change in CAT with 80% power, 60 people per group are required (α=0.05, two tailed). In anticipation of a possible 25% attrition rate, the total sample size was increased to 75 per group (150 in total).

### Quantitative analysis
This will primarily be completed on an intention-to-treat analysis. All quantitative data will be assessed for normality and analysed using appropriate parametric and non-parametric statistics (eg, within and between measures t-tests and analyses of variance); statistical significance will be set at p=0.05. Secondary per protocol analyses will be carried out.

A post hoc analysis will be carried out, which will exclude patients in either arm that received pulmonary rehabilitation as part of their usual care. We would anticipate that patients who participate within the study will not require pulmonary rehabilitation. However, due to pulmonary rehabilitation being a part of 'best' usual care, this will not be withheld from the patient.

Quantitative data for all outcomes will be transcribed from the case report form (CRF) onto an electronic database. A statistical software package will be used to carry out quantitative analyses. Predictive analytics software (SPSS; Statistical Package for the Social Sciences) will be used to analyse the data, the licence for which is provided by University Hospitals of Leicester NHS Trust. Continuous variables will be presented as mean and SD or median and IQR, and categorical data will be presented as frequencies and percentages. Data will be checked for normality and appropriate parametric and non-parametric tests will be used. Any baseline differences will be adjusted for. Any missing data will be imputed, and both intention-to-treat and per-protocol analyses will be conducted.

We have not secured funding for a healthcare utilisation analysis but would anticipate further CLAHRC (Collaboration and Leadership in Applied Health Research and Care) funding if the trial is clinically effective.

### Qualitative analysis
The focus groups will be analysed using Thematic Analysis[32] supported by NVivo software (V.9). This approach follows six distinct stages: familiarisation with data, generating initial codes, searching for themes, reviewing themes, defining and naming themes; and producing the report. The psychologist and the physiotherapist will carry out initial coding and a sample of interviews will be coded by another member of the team to ensure consistency and to enhance interpretive authenticity. Throughout the data analysis, the team will meet to discuss and review emerging themes and search for accounts that provide contesting views of the same phenomena or identify different phenomena. Our patient representatives will be invited to comment on our (anonymised) findings throughout the analysis process to ensure interpretations made by researchers stay close to the direct experience of patients.[33]

All patient information that is collected during the course of the research will be kept strictly confidential. Any information about the patient who leaves the hospital will have their name and address removed. Participants will not be identified in any subsequent written material. Results will be reported in such a way that completely preserves confidentiality.

## ETHICS AND DISSEMINATION
### Ethics
The trial is sponsored by the University Hospitals of Leicester NHS Trust (study number 152408), and ethical approval was granted by the Hampshire B Research Ethics Committee (REC reference: 14/SC/1169). Protocol amendments will be approved by the ethics committee and regulatory authorities as per current guidelines and will be communicated to relevant parties by the study team.

### Dissemination
We plan to publish the results of the study in peer-reviewed journals and present them at appropriate national and international respiratory and physiotherapy conferences. Social media will be used to disseminate information and summaries of results to a wider population.

The CLAHRC East Midlands is a large organisation that strives to improve health outcomes in the population across the East Midlands through delivering high-quality, world class research. This organisation will be used to further disseminate results within the East Midlands. We also hope to provide a summary of results to the study participants. Furthermore, we plan to hold a participant dissemination day towards the end of the study. This will enable participants to contribute to the final report and other result dissemination activities.

The institution also has an active and dynamic public involvement group for pulmonary and cardiovascular rehabilitation. The group will be used to create and coordinate strategies for further disseminating the results into the public domain.

The study may also be subject to internal and further external audits to ensure safety of the trial.

## CONCLUSION
The importance of self-management is widely acknowledged, and opportunities should be maximised from the time of diagnosis through to more severe disease. Opportunities to improve self-management skills should be embedded in a pulmonary rehabilitation programme. In the future, there may be an opportunity to explore the value of the SPACE for COPD(C) programme alongside rehabilitation, or indeed, an alternative for those unwilling or unable to attend. However, for those with milder disease, there is no provision for a structured supported self-management programme in the UK. Evidence suggests that the SPACE for COPD(C) package is effective when delivered on an individual basis.[10] This study examines its effectiveness as a group-based intervention in the community, as an alternative supported self-management strategy, which importantly allows patient choice.

### Protocol version
11 18.11.2015. Study started on 02/2015, and ends in 06/2017. Recruitment was 20 months.

**Acknowledgements** East Leicestershire CCG: Sue Price; Leicestershire Partnership Trust: Karen Moore, Alex Woodward and Gillian Doe; West Leicestershire CCG: Jake Cooke; Public Involvement: Patricia Overty and Freda Smart.

**Contributors** SS is the principal investigator of the SPACE for COPD(C) study. KM, SS, SSc, LA, LH-W, SH and CB were involved in the development of the intervention and design of the trial. CB, SS, PK and TH-D have been involved in drafting the work or revising it critically for important intellectual content and have given the final approval of the version published.

**Funding** The research is funded by the National Institute for Health Research (NIHR) Collaboration for Leadership in Applied Health Research and Care East Midlands (CLAHRC EM) and took place at the University Hospitals of Leicester NHS Trust. Support is also provided by the NIHR Leicester Respiratory Biomedical Research Unit (BRU). The views expressed are those of the authors and not necessarily those of the NHS, the NIHR or the Department of Health. Study sponsor, Carolyn Maloney, University Hospitals of Leicester NHS Trust, 0116 258 4109.

**Disclaimer** Roles of the funder: management of staff, study progress, reporting and dissemination activities.

**Competing interests** None declared.

**Ethics approval** Hampshire B Research Ethics Committee.

**Provenance and peer review** Not commissioned; externally peer reviewed.

**Data sharing statement** Additional unpublished data from the study is still being collected and analysed and is only available to members of the study team.

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
