## [Reviewer comments · BMJ Open]

ARTICLE DETAILS

TITLE (PROVISIONAL)	A self-management programme of activity coping and education - SPACE FOR COPD© - in primary care: The protocol for a pragmatic trial
AUTHORS	Bourne, Claire; Kanabar, Pratiksha; Mitchell, Katy; Schreder, Sally; Houchen-Wolloff, Linzy; Bankart, M. John; Apps, Lindsay; Hewitt, Stacey; Harvey-Dunstan, Theresa; Singh, Sally

VERSION 1 - REVIEW

REVIEWER	Suzanne C Lareau University of Colorado, Denver, Colorado, USA
REVIEW RETURNED	05-Oct-2016

GENERAL COMMENTS	The focus of the document was more systematic on evaluating outcomes related to the patient than the HCP. Might consider a more systematic evaluation of the providers and staff who are presumably administering the SPACE program. It is unclear why the CAT (a predominantly symptom assessment) was chosen as the primary outcome measure, since the RCT showed dyspnea was not significant between treatment and control group. Over what period of time will the study be offered in the community? Can HCP's opt out of this program? References # 3 and 4 incomplete
--

REVIEWER	Brenda O'Neill Centre for Health and Rehabilitation Research, Institute Nursing and Health Research, Sch Health Sciences, Ulster University, Northern Ireland
REVIEW RETURNED	21-Oct-2016

GENERAL COMMENTS	This protocol outlines a study which aims to explore feasibility, acceptability and efficacy of an intervention [SPACE] delivered by HCP to groups of patients with COPD rather than researchers and experienced clinicians; and to examine whether group based delivery of SPACE FOR COPD, with sustained support, improves the maintenance of outcomes following the SPACE FOR COPD intervention. Importantly the study appears to support an intervention that will offer patients with COPD an opportunity to take an active role in self-managing their condition, by delivering sessions across a number of
---

months, and which will be held at community venues with patients mostly in primary care. Clearly there is a huge amount of preparation involved for execution of this study.

Abstract and overall aims

The aims are presented at the end of the abstract. It would be helpful if the overall aims were additionally presented in the main text, just prior to the specific study objectives.

Minor point - does this bit “rather than researchers and experienced clinicians” need to be in the aims, as the study does not compare delivery via HCPs to delivery via researchers?

Introduction

An overview of relevant literature is provided. Objective 1 “and to examine whether group-based delivery of SPACE FOP COPD with sustained support improves the maintenance of outcomes following the SPACE FOP COPD intervention” – consider adding ‘compared to a control group’. The wording relating to “improves the maintenance of outcomes following the intervention” might need some clarification. Is this referring to whether 6mth improvements are sustained at 9mths?

Methods

The methods are presented in detail and the overall protocol aligns with the SPIRIT checklist. There are some points for consideration and/or clarification below:

The proposed group size is up to 10 per group. Is there a minimum group size for this intervention to be considered “group based” delivery or is group size an aspect of feasibility that will be explored/captured?

The detail of the SPACE FOR COPD is well articulated and tabulated in the protocol. I couldn’t quite see if there was any support offered outside of the 6 face to face group sessions but might have missed this.

The methods for exploring aspects of intervention fidelity are summarised and include the use of check lists and observation. There is a summary of the study OMs included.

Overall there could be more detail on the aspects of feasibility and acceptability that are to be examined i.e. aspects of feasibility that will be considered within the RCT trial e.g. retention and adherence, and aspects of feasibility and acceptability from HCP and from the patients perspective.

Study power: Is the study powered to detect short term change/effectiveness i.e. change at 6mths or change at 9 mths?

For the qualitative component “Semi structured focus groups” please clarify the terminology for the qualitative method e.g. focus groups or semi structured interviews. With regards the sampling, is the intent to include completers and non-completers in the qualitative component as this could provide useful insight. The method for analysis is otherwise outlined.

Other

Figure 1 box 1- Should the order be: screening, consent, assessment. It appears there is concealed allocation and if so, this

	could be added to the methods. Consider a short discussion/conclusion for this protocol. For example perhaps consider including whether self management is currently fully captured within the context of pulmonary rehabilitation, and how the SPACE fits with practice relating to pulmonary rehabilitation and overall management of COPD. Please check the tense in the protocol e.g. P 17 'registered health professions were trained', P18 'data is collected' vs will be, p20 see Data sharing statement exercise capacity 'was' measured p 29. Thanks you for the opportunity to review this protocol in an important clinical area.
--	--

REVIEWER	Kjersti Grønning Norwegian University of Science and Technology Norway
REVIEW RETURNED	23-Jan-2017

GENERAL COMMENTS	The study is very interesting. However, I have a few remarks I would like you to answer/ /elaborate (from the review checklist): 4. Are the methods described sufficiently to allow the study to be repeated?  • Could you be more precise regarding comorbidities" (page 10) and the eligibility criteria (exclusion criteria). Do you mean any other disease in addition to COPD? • Table 3: At 6 months you use a blinded study team member in the data collection (very good). Why do you allow an un-blinded study team member in the data collection at 9 months? • Qualitative Data Collection: I would like you to elaborate why you have chosen focus groups as most appropriate. You also need to argue why you need up to 10 focus groups? Focus groups usually consist of 6-10 participants; why do you allow only 4? (page 23) • You will invite the ones delivering the intervention to a meeting to discuss and gain insight into any potential facilitators/barriers to implementing the intervention. Why not conduct a focus group interview for the HP as well? (page 23) 7. If statistics are used are they appropriate and described fully?  • Study Power: Is the change in CAT a clinically meaningful difference? (page 24) • Could you be more precise in what kind of statistics you consider is appropriate parametric and non-parametric statistics? (page 25) • Why not invite your patient representatives to participate in the whole qualitative analyses process and not only comment on your initial findings? 8. Are the references up-to-date and appropriate?  • I am not able to locate reference number 4, please check if it is
--

	correct. “Purdy S. Avoiding hospital admissions: What does the research evidence say? 2010.” • Year of publication is missing in ref number 5.
--	--

VERSION 1 – AUTHOR RESPONSE

Reviewer: 1

Comment 1: “The focus of the document was more systematic on evaluating outcomes related to the patient than the HCP. Might consider a more systematic evaluation of the providers and staff who are presumably administering the SPACE program.”

Response to comment 1: We thank the reviewer for this suggestion. The main focus of the study was to understand whether the intervention works in the new format in regards to patient outcomes. As there will only be a small number of HCPs delivering the intervention we feel that a group discussion allows us to understand their thoughts on whether this was acceptable/feasible and how they think the intervention might work in practice. We aim to generate responses of a more practical nature, therefore, full thematic analysis is not required thus removing the need for transcription and audio recording. Discussions will be conducted with the rigour of focus groups.

Comment 2: “It is unclear why the CAT (a predominantly symptom assessment) was chosen as the primary outcome measure, since the RCT showed dyspnea was not significant between treatment and control group.”

Response to comment 2: Thank you for this comment. As this is a pragmatic trial we are trying to understand any difference in COPD symptoms using a measure that will be easy to use in clinical practice. Additionally, in the previous RCT change in dyspnoea was significant post intervention. Within this trial, the 6-month appointment can be classed as post-intervention due to the length of time the intervention lasts (approximately 5 months). We have added the following sentence to the primary outcome paragraph to address this comment (page 21):

‘This measure was chosen due to ease of use in clinical practice compared to the Chronic Respiratory Questionnaire (as used in the previous RCT).’

Comment 3: “Over what period of time will the study be offered in the community?”

Response to comment 3: The study will be offered over a period of 2 years (the duration of the active component of the intervention within the study) and will only be offered as part of the research, not routine practice within the community. We have added the following to the intervention paragraph (pages 12 and 13):

‘Participants will also be asked to complete the exercise component of the manual at home in their own time. A full description of the rationale, development and efficacy of the work underpinning the SPACE for COPD© manual is detailed elsewhere (9). The intervention will be offered over a period of 2 years (the duration of the active component of the intervention within the study) and will only be offered as part of the research, not routine practice within the community.’

Comment 4: “Can HCP's opt out of this program?”

Response to comment 4: Yes, a collaborative agreement was made with the community COPD team as to whose time is allocated to delivering the intervention; however, if any of these HCPs are unable to do so then cover will be provided. Additionally, some hospital staff were trained to deliver the intervention and they are involved in delivering some of the groups and providing cover. Only those who wished to be trained were.

Comment 5: "References # 3 and 4 incomplete"

Response to comment 5: Thank you for spotting this error, this has now been rectified. The references now read (page 34):

(3) NHS England. Five year forward view. October 2014. (report) <https://www.england.nhs.uk/wp-content/uploads/2014/10/5yfv-web.pdf>

(4) Purdy S. Avoiding hospital admissions: What does the research evidence say? 2010. The King's Fund (report). <https://www.kingsfund.org.uk/sites/files/kf/Avoiding-Hospital-Admissions-Sarah-Purdy-December2010.pdf>

Reviewer: 2

Thank you for your complementary summary of our protocol.

Abstract and overall aims

Comment 1: "The aims are presented at the end of the abstract. It would be helpful if the overall aims were additionally presented in the main text, just prior to the specific study objectives."

Comment 2: "Minor point - does this bit "rather than researchers and experienced clinicians" need to be in the aims, as the study does not compare delivery via HCPs to delivery via researchers?"

Response to comments 1 and 2: These two points have been addressed. We felt that adding the aims before the objectives sounded repetitive and upon further consideration saw bullet points one and two as aims, and points 3 and 4 as the study objectives that should come under the second aim.

Therefore, the study aims and objectives now read (page 8):

1. To examine whether group-based delivery of SPACE FOR COPD©, with sustained support, improves outcomes following the intervention compared to a control group.
2. To explore feasibility, acceptability and efficacy of the intervention to be delivered and supported by HCP's. This will be done by:
 - a. Exploring HCP's experiences of delivering the intervention and identify any barriers to delivery in practice.
 - b. Understanding, from the patient perspective, the feasibility and acceptability of the SPACE for COPD© intervention delivered by HCPs in a group-based, community setting.

Introduction

Comment 3: An overview of relevant literature is provided. Objective 1 "and to examine whether group-based delivery of SPACE FOP COPD with sustained support improves the maintenance of outcomes following the SPACE FOP COPD intervention" – consider adding 'compared to a control group'.

Response to comment 3: Thank you, we agree with this comment. We have now added this to the

protocol and the objective (aim) now reads (page 8):

1. To examine whether group-based delivery of SPACE FOR COPD©, with sustained support, improves patient outcomes following the intervention compared to a control group.

Comment 4: “The wording relating to “improves the maintenance of outcomes following the intervention” might need some clarification. Is this referring to whether 6mth improvements are sustained at 9mths?”

Response to comment 4: We agree that this wording is not as intended and as such have taken out ‘maintenance of outcomes’. That aim now reads (page 8):

1. To examine whether group-based delivery of SPACE FOR COPD©, with sustained support, improves patient outcomes following the intervention compared to a control group.

Methods

The methods are presented in detail and the overall protocol aligns with the SPIRIT checklist. There are some points for consideration and/or clarification below:

Comment 5: “The proposed group size is up to 10 per group. Is there a minimum group size for this intervention to be considered “group based” delivery or is group size an aspect of feasibility that will be explored/captured?”

Response to comment 5: Thank you for this comment. Group size will be an aspect of feasibility that will be explored/captured. We have addressed this as part of your comment below (comment 7) about additional detailing of feasibility/acceptability measures.

Comment 6: “The detail of the SPACE FOR COPD is well articulated and tabulated in the protocol. I couldn’t quite see if there was any support offered outside of the 6 face to face group sessions but might have missed this.”

Response to comment 6: Thank you for this comment and for highlighting that we have not included this. There is no additional structured support offered to participants outside of the 6 face-to-face sessions although participants are provided with a contact number for at least one of the programme facilitators should they have any queries/are unable to attend any sessions. We have therefore added an additional sentence at the end of the method section for the intervention (page 12):

‘Participants are provided with a contact number for at least one of the facilitators throughout the course of the programme in case they have any further queries/are unable to attend any sessions.’

Comment 7: “Overall there could be more detail on the aspects of feasibility and acceptability that are to be examined i.e. aspects of feasibility that will be considered within the RCT trial e.g. retention and adherence, and aspects of feasibility and acceptability from HCP and from the patients perspective.”

Response to comment 7: We agree and have now addressed this comment and added the following to the outcomes section (pages 24-25):

‘Outcomes to assess feasibility and acceptability of trial parameters

We will use the following outcomes to assess the feasibility and acceptability of trial parameters:

Screening

Defined as the number of packs sent out to patients from GP practices and assessed for eligibility using inclusion/exclusion criteria by a study researcher.

Eligibility

Calculated by dividing number of people screened for eligibility by those who meet the inclusion criteria.

Consent

Defined as the proportion of people with COPD who met inclusion criteria, and were therefore eligible, who went on to consent in writing to participate in the study.

Group characteristics

Group characteristics (e.g., age, gender, GOLD stage, MRC, exercise capacity, physical activity) will be compared between completers and non-completers.

Retention

Defined as the number of participants who remain in the study and do not drop-out.

Study Completion

Defined by the number of participants who complete the COPD Assessment Test. Completion rates will be calculated at baseline, 6-month and 9-month follow-up.

Intervention adherence and completion rates

This will be measured by summing the total number of self-management programme sessions attended by participants allocated to the intervention group. We will also look at the average group size across each of the six self-management programme sessions and compare with the number of participants allocated to each of the self-management programme groups.'

- We also feel that discussing potential facilitators/barriers to implementing the intervention in practice (and derive practical recommendations for doing so) with HCPs encompasses feasibility and acceptability aspects. We have re-worded this section to better reflect this (page 27):

'Healthcare professionals delivering the SPACE for COPD© self-management support intervention will be invited to participate in a meeting to discuss aspects of feasibility and acceptability, such as gaining insight into any potential facilitators/barriers to implementing the intervention in practice (and derive practical recommendations for doing so).'

Comment 8: "Study power: Is the study powered to detect short term change/effectiveness i.e. change at 6mths or change at 9 mths?"

Response to comment 8: The study is powered to detect change in the primary outcome at 6-months. This is outlined in the power calculation on page 27.

Comment 9: "For the qualitative component "Semi structured focus groups" please clarify the terminology for the qualitative method e.g. focus groups or semi structured interviews. With regards the sampling, is the intent to include completers and non-completers in the qualitative component as

this could provide useful insight.”

Response to comment 9: Thank you for highlighting these points. The qualitative method used is focus groups. We have removed ‘semi-structured’ from the protocol to reflect this. With regards to sampling, the intention is to include those that attend the final self-management session as focus groups will be carried out at the end of this session. This methodology was discussed with our patient representatives who were mindful of asking participants coming back for an additional time when it would be easier for them to stay longer in the last self-management session. This may mean that those who are not classed as completers of the intervention, but attend the last session, are still included in the focus group. Every attempt will be made to remind participants of this occurring during the final session regardless of how many other sessions have been attempted.

The following has been added to the protocol (page 26):

‘Focus groups will be carried out at the end of the last group session for participant ease, as discussed with study patient representatives. We will prompt participants allocated to each self-management group of the focus group discussion prior to the last session in the attempt to gain experiences from as many participants as possible, regardless of the number of sessions attended in total.’

Other

Comment 10: Figure 1 box 1- Should the order be: screening, consent, assessment. It appears there is concealed allocation and if so, this could be added to the methods.

Response to comment 10: We agree that the order has been inputted incorrectly and has now been corrected in figure 1 (page 33). We believe we have provided all allocation details within table 3: Details of study clinical assessments and outcome measures at all appointments (page 20).

Comment 11: “Consider a short discussion/conclusion for this protocol. For example perhaps consider including whether self management is currently fully captured within the context of pulmonary rehabilitation, and how the SPACE fits with practice relating to pulmonary rehabilitation and overall management of COPD.”

Response to comment 11: Thank you for your suggestion. We have added the following to the end of the protocol (page 31):

‘CONCLUSION

The importance of self-management is widely acknowledged and opportunities should be maximised from the time of diagnosis through to more severe disease. Opportunities to improve self-management skills should be embedded in a pulmonary rehabilitation programme. In the future, there may be an opportunity to explore the value of the SPACE for COPD© programme alongside rehabilitation, or indeed, an alternative for those unwilling or unable to attend. However, for those with milder disease there is no provision for a structured supported self-management programme in the UK. Evidence suggests that the SPACE for COPD© package is effective when delivered on an individual basis (10). This study examines its effectiveness as a group-based intervention in the community, as an alternative supported self-management strategy, which importantly allows patient choice.’

Comment 12: “Please check the tense in the protocol e.g. P 17 ‘registered health professions were trained’, P18 ‘data is collected’ vs will be, p20 see Data sharing statement exercise capacity ‘was’

measured p 29.”

Response to comment 12: These instances of incorrect tense have all been addressed and those sentences now read:

Pg 17 (now 18): In total, 8 registered health professionals will be trained to deliver the SPACE for COPD© group-based intervention by two Health Psychologists.

Pg 18 (now 19): Data will be collected during baseline, 6-month and 9-month appointments at the Leicester Respiratory Biomedical Research Unit by trained members of the study team.

Pg 20 (now 21): Endurance capacity will be measured with the endurance shuttle walk test (ESWT) using a 10-m course and a walking speed of 85% of the maximal ISWT walking speed (19).

Pg 29 (now 31): The research is funded by the National Institute for Health Research (NIHR) Collaboration for Leadership in Applied Health Research and Care East Midlands (CLAHRC EM), and took place at the University Hospitals of Leicester NHS Trust. Support is also provided by the NIHR Leicester Respiratory Biomedical Research Unit (BRU). The views expressed are those of the authors and not necessarily those of the NHS, the NIHR or the Department of Health.

Reviewer: 3

Comment 1: “Could you be more precise regarding comorbidities” (page 10) and the eligibility criteria (exclusion criteria). Do you mean any other disease in addition to COPD?”

Response to comment 1: Thank you for highlighting this ambiguous wording. We have now further clarified the point and the exclusion criteria now read (page 10):

‘Patients are excluded from participating in the trial if they are:

- Unable to participate in the exercise component of the SPACE for COPD© programme due to neurological, locomotive, or psychiatric disability
- Unable to participate in the exercise component of the SPACE for COPD© programme due to other comorbidities where exercise would be a contraindication (for example, unstable angina).
- Unable to read/write English to the level of an eight year old
- Unwilling to be randomised
- Previous participants of Pulmonary Rehabilitation or have received the SPACE for COPD© manual in the previous 12 months.’

Comment 2: “Table 3: At 6 months you use a blinded study team member in the data collection (very good). Why do you allow an un-blinded study team member in the data collection at 9 months?”

Response to comment 2: We thank you for your comment. Unfortunately, practical barriers such as cost and personnel would not allow for the additional support needed for a blinded study team member to complete all 6 and 9-month data collection measures.

Comment 3: “Qualitative Data Collection: I would like you to elaborate why you have chosen focus groups as most appropriate. You also need to argue why you need up to 10 focus groups? Focus groups usually consist of 6-10 participants; why do you allow only 4? (page 23)”

Response to comment 3: Thank you for this comment and have elaborated on our decision processes.

As we are trying to understand whether group-based delivery of the SPACE for COPD© manual is

effective we chose focus groups due to their generation of information on collective views, and the meanings that lie behind those views. They are also useful in generating a rich understanding of participants' experiences and beliefs. We are conducting focus groups with participants who are members of each self-management programme group as pre-existing groups have shared experiences and enjoy a comfort and familiarity which facilitates discussion or the ability to challenge each other comfortably. As such, focus groups will be conducted with every self-management group to understand experiences of groups from different areas. We will also allow for small numbers to allow for drop-out.

We have explained this more fully in the protocol and the section now reads (pages 25-26):

'Those participants allocated to the intervention group are invited to take part in qualitative focus groups at the end of the SPACE for COPD© intervention. Focus groups have been chosen due to their generation of information on collective views, and the meanings that lie behind those views. The aim of the focus groups will be to understand participants' experiences of the group-based self-management programme. More specifically, the data we collect will inform:

- Acceptability and usefulness of the programme to participants in this format and over this time period;
- The content of the intervention;
- Approaches to recruitment.

Focus groups will be conducted with each self-management programme group, with between 3 and 10 participants (number dependent on each group size). This difference in participant numbers allows for participant opinions to be gathered even if a small group is encountered (e.g., due to drop-out). Although three is a very small number for a focus group, it allows all participant opinions to be gathered, regardless of group size. Participants will be familiar with one another (which can help facilitate discussion or the ability to challenge each other comfortably) as they have attended multiple group sessions together.'

Comment 4: "You will invite the ones delivering the intervention to a meeting to discuss and gain insight into any potential facilitators/barriers to implementing the intervention. Why not conduct a focus group interview for the HP as well? (page 23)"

Response to comment 4: The purpose of the HCP discussion is to understand the practicalities and feasibility of delivering the intervention in practice. As we aim to generate responses of a more practical nature, full thematic analysis is not required thus removing the need for transcription and audio recording. Discussions will be conducted with the rigour of focus groups.

Comment 5: "Study Power: Is the change in CAT a clinically meaningful difference? (page 24)"

Response to comment 5: Yes, this change in CAT is a clinically meaningful difference, as outlined by Kon et al., 2014 (Kon SS, Canavan JL, Jones SE, Nolan CM, Clark AL, Dickson MJ, Haselden BM, Polkey MI & Man WD. 2014. Minimum clinically important difference for the COPD Assessment Test: a prospective analysis, *Lancet Respir Med* 2(3):195-203). Kon et al., state that "the most reliable estimate of the minimum important difference of the CAT is 2 points."

Comment 6: "Could you be more precise in what kind of statistics you consider is appropriate parametric and non-parametric statistics? (page 25)"

Response to comment 6: We have added in another sentence providing more detail to address this comment which reads (page 27):

'All quantitative data will be assessed for normality and analysed using appropriate parametric and non-parametric statistics (e.g., within and between measures t-tests and ANOVAs), alpha will be set at 0.05. Secondary per protocol analyses will be carried out.'

Comment 7: "Why not invite your patient representatives to participate in the whole qualitative analyses process and not only comment on your initial findings?"

Response to comment 7: We agree that this is an oversight and after speaking to our patient representatives have now included this in the protocol (page 29):

'Our patient representatives will be invited to comment on our (anonymised) findings throughout the analysis process to ensure interpretations made by researchers stay close to the direct experience of patients (33).'

Comment 8: "I am not able to locate reference number 4, please check if it is correct. "Purdy S. Avoiding hospital admissions: What does the research evidence say? 2010."

Comment 9: "Year of publication is missing in ref number 5."

Response to comments 8 and 9: Thank you for highlighting these oversights. These have now been fully completed. They now read (page 34):

(4) Purdy S. Avoiding hospital admissions: What does the research evidence say? 2010. The King's Fund (report). <https://www.kingsfund.org.uk/sites/files/kf/Avoiding-Hospital-Admissions-Sarah-Purdy-December2010.pdf>

(5) Zwerink M, Brusse-Keizer M, van der Valk PD, et al. Self management for patients with chronic obstructive pulmonary disease. Cochrane Database of Systematic Reviews. 2014.

We would like to thank you and your reviewers for their comments. We hope our revised manuscript now meets with your approval. Please feel free to contact me if you require any further clarification.

VERSION 2 – REVIEW

REVIEWER	Suzanne C Lareau University of Colorado, Anschutz Medical Campus, Aurora, CO, USA
REVIEW RETURNED	24-Feb-2017

GENERAL COMMENTS	Well done!
------------